# Salivary Enzymatic Activity and Carious Experience in Children: A Cross-Sectional Study

**DOI:** 10.3390/children9030343

**Published:** 2022-03-02

**Authors:** Raluca-Paula Vacaru, Andreea Cristiana Didilescu, Ileana Constantinescu, Ion Mărunțelu, Mihaela Tănase, Ioana Andreea Stanciu, Wendy Esmeralda Kaman, Hendrik Simon Brand

**Affiliations:** 1Division of Embryology, Faculty of Dental Medicine, Carol Davila University of Medicine and Pharmacy, 8 Eroii Sanitari Boulevard, 050474 Bucharest, Romania; raluca.vacaru@umfcd.ro; 2Division of Pedodontics, Faculty of Dental Medicine, Carol Davila University of Medicine and Pharmacy, 8 Barajul Iezeru, 032799 Bucharest, Romania; mihaela.tanase@umfcd.ro (M.T.); ioana.stanciu@umfcd.ro (I.A.S.); 3Centre for Immunogenetics and Virology, Fundeni Clinical Institute, Carol Davila University of Medicine and Pharmacy, 258 Fundeni Road, 022328 Bucharest, Romania; ileana.constantinescu@umfcd.ro (I.C.); ion.maruntelu@drd.umfcd.ro (I.M.); 4Department of Oral Biochemistry, Academic Centre for Dentistry Amsterdam (ACTA), VU University of Amsterdam and University of Amsterdam, Gustav Mahlerlaan 3004, 1081 LA Amsterdam, The Netherlands; w.e.kaman@acta.nl (W.E.K.); h.brand@acta.nl (H.S.B.)

**Keywords:** dental caries, children, saliva, alpha-amylase, total protease activity, matrix metalloproteinase

## Abstract

Salivary biomolecules are considered important modulators of the oral microflora, with a potential subsequent impact on dental health. The present study aimed to investigate the relationship between salivary enzymatic activity and carious experience in children. The carious experience of a sample of 22 school children was evaluated by calculating dmf/DMF indices, following WHO recommendations. Unstimulated whole saliva was collected, and salivary alpha-amylase levels, total protease activity, and matrix metalloproteinase levels (MMP-8 and MMP-9) were measured. The data were analyzed using parametric and nonparametric tests. Our findings revealed no significant relationship between the investigated salivary parameters and the carious experience in permanent teeth (DMFT/DMFS scores). Carious indices scores for primary teeth (dmft and dmfs) were positively associated with MMP-8 levels (*r* = 0.62, *p* = 0.004 and r_s_ = 0.61, *p* = 0.006, respectively) and MMP-9 levels (*r* = 0.45, *p* = 0.05 and *r_s_* = 0.48, *p* = 0.039, respectively) and negatively associated with alpha-amylase levels (*r_s_* = −0.54, *p* = 0.017 and *r_s_* = −0.59, *p* = 0.006, respectively). Although with a marginal significance, PEK−054 levels positively correlated with dental caries, while for PFU−089, a negative correlation was observed. These results suggest that salivary alpha-amylase and MMP-8 and MMP-9 levels may be considered potential indicators of carious experience in children. Further studies with a prospective design are needed in order to elucidate the role of these biomolecules in caries development.

## 1. Introduction

Regardless of the abundance of resources available, as well as worldwide agreed-upon recommendations and guidelines, dental caries is still a major public health challenge. It remains one of the most prevalent non-communicable chronic diseases, despite being completely preventable and manageable [1,2,3]. Dental caries affects all age categories and has a major impact on an individual’s quality of life and general health. Children, in particular, are directly affected by the consequences of untreated caries, often complicated by the loss of pulp vitality, which can have functional, aesthetic, physical, and psychological impacts [4]. Due to pain and the loss of masticatory units, children might have impaired feeding ability, which consequently negatively affects their physical development. Furthermore, it has been shown that children affected by dental caries are prone to have poorer school attendance rates and performances, as well as compromised social interactions [5].

In order to avoid these consequences, it is necessary to understand the mechanism of occurrence and development of dental caries, as well as to identify the contributing factors. Dental caries is a disease with multifactorial etiology, which includes endogenous and exogenous factors. For a carious lesion to develop, it is necessary that a prolonged action of cariogenic bacteria, such as *Streptococcus mutans* and *Lactobacilli* spp., on a fermentable carbohydrate substrate occurs [6], favored by imbalanced host conditions in relation to dietary habits, oral hygiene, quantity and composition of an individual’s saliva, and dental structure. These factors may subsequently modify the dynamics of caries progression [7]. Deep dentin caries in permanent teeth is characterized by higher counts of *Lactobacilli*, while *S. mutans* is dominant in primary teeth lesions [8,9,10,11].

Saliva is a biological fluid secreted by the salivary glands with a complex composition, comprising both organic and inorganic elements. Saliva has a major role in oral physiology and, under normal conditions, performs an adequate elimination of cariogenic alimentary substrates from tooth surface. Moreover, because of its buffering and remineralizing actions, saliva acts as a protector against dental caries development and progression [12]. In recent studies, some salivary proteins have been recognized as indicators of dental caries, but limited data are available regarding these associations [13,14,15]. Therefore, potential biomarkers need to be studied more specifically in order to gain insight in how the oral environment influences the occurrence of dental caries. Investigating enzyme activity might offer a broad image of the links established at the molecular level [16].

An important salivary enzyme studied in association with dental caries, which has various biological implications, is alpha-amylase (sAA). sAA is a digestive calcium-dependent metalloenzyme, synthesized in the acinar cells of the salivary glands. It is involved in the hydrolyzation of large polysaccharides, e.g., in the degradation of starch molecules into dextrin and subsequently into maltose and glucose [17,18]. Besides its digestive function, sAA plays various other roles in salivary physiology. sAA has been recognized for its anti-bacterial functions by regulating oral bacterial colonization and by favoring bacterial agglutination, hereby facilitating their elimination [19,20,21]. On the other hand, by binding to cariogenic bacteria and to the tooth surface, sAA is incorporated in the oral biofilm where it facilitates the provision of glucose, the substrate for acid production and subsequent demineralization of dental structures. sAA attaches to bacterial plaques through amylase-binding bacteria and provides the necessary glucose for bacterial metabolism by hydrolyzing alimentary starch adjacent to the teeth’s surface [19].

A bacterial acidic attack alone is not sufficient for the development of a carious cavity within the dental structure, because it only affects the mineral structure through demineralization. Although the destruction of organic matrix was firstly thought to be attributed to bacterial proteases [22,23,24,25], this could not be demonstrated in vitro, which roused the idea of host-derived MMPs being implicated [26]. Various studies have shown that, in addition to a bacterial acidic attack, the organic matrix of the tooth is disintegrated under the action of endogenous proteolytic enzymes, such as matrix metallo-proteinases (MMPs) [26,27]. MMPs are a group of endopeptidase enzymes that disintegrate the extracellular matrix (ECM) and other bioactive molecules, such as cytokines and chemokines. Twenty-four genes encoding MMPs have been identified and characterized in humans [28]. During tooth morphogenesis, odontoblasts produce MMPs that have a substantial role in dentin formation and afterwards remain in an inactive form, entrapped in the dentinal structure [29]. When subjected to low pH, such as during bacterial acidic attack, these endogenous pro-MMPs are activated, locally exerting their proteolytic activity [30]. The released MMPs reach the saliva. On the other hand, not only dentin-bound MMPs are responsible for ECM degradation during the caries process but also MMPs originating from oral fluids that can penetrate through the enamel–dentin complex and become involved in the destructive mechanism [31]. This occurs especially in immature teeth that exhibit larger dentinal tubules and in teeth with affected structure [32]. In the oral cavity, MMPs are associated with periodontal disease, dental caries, and pulp inflammation. In whole saliva, collagenases (MMP-8) and gelatinases (MMP-2 and MMP-9) are present and are thought to originate mainly from gingival crevicular fluid and salivary glands’ secretions. Their activity is associated with the condition of the periodontium. Recently, the focus of MMP-related research shifted to the study of the role of MMPs in the development of carious disease with the goal to understand whether MMPs can serve as biomarkers in the diagnosis of carious disease. Some arguments that support the involvement of salivary MMPs in caries pathogenesis are the facts that the exposure of tooth structure to external collagenase leads to cavitation [33] and that locally applied MMP inhibitors can reduce dentin caries progression [28]. However, the current available data are inconclusive. Among the MMPs detected in dentin (MMP-2, -8, -9, -14, and -20), MMP-8 is the highest contributor to the degradation of type I collagen, which represents the bulk protein in dentin [34,35]. Nonetheless, MMPs represent only a small fraction of total proteolytic activity and may either originate from the host or have a bacterial origin. Therefore, analyzing the role of MMPs in caries onset in a broader context might unveil interesting results.

Accordingly, the main objective of the present study was to elucidate the relationship between salivary enzymatic activity and carious experience in children. In addition, we aimed to investigate whether the salivary levels of these biomolecules reflect disease severity.

## 2. Materials and Methods

### 2.1. Design of the Study

This was a cross-sectional study carried out on 22 healthy school children, who visited the Pediatric Dentistry Clinic of the Faculty of Dental Medicine, Carol Davila University of Medicine and Pharmacy, Bucharest, for dental treatments between 2019 and 2020. This study was approved by the Research Ethics Committee at Carol Davila University of Medicine and Pharmacy (no. 188/28.01.2019). Before enrolment, a written informed consent was signed by the parents or legal guardians of the participants after they received information about the aims and protocols of the study. The subjects were submitted to the study if they presented at least one carious lesion. Participants with a history of systemic diseases or mental disabilities and those who received medical treatments in the last 3 months that might have affected salivary gland functioning were excluded. 

### 2.2. Clinical Examinations

Standard clinical examinations were performed by a single trained examiner (RPV) using a dental mirror and an explorer, in a dental office under adequate artificial light conditions and following World Health Organization recommendations [36]. The examinations were initiated from the last tooth on the right maxillary arch, following through the left maxillary, left mandibular, and right mandibular arches. Dental caries was diagnosed through visual inspection and probing, on both dry and wet surfaces. Caries status was recorded as DMF/dmf indices, for teeth and surfaces, respectively. Initial-stage caries was not considered for the calculation of the DMF/dmf indices. The indices recorded for primary dentition were dmft/dmfs, while those for permanent dentition were DMFT/DMFS; for mixed dentition, both scores were recorded separately. The values of the indices were obtained by summing the decayed (D/d), missing due to caries (M/m), and filled due to caries (F/f) scores. Furthermore, for a global evaluation of the carious experience, we defined MCL (manifest caries lesions) as the number of teeth affected by untreated carious lesions and we also calculated the percentage of teeth affected by untreated caries from the total number of teeth present in the oral cavity. Furthermore, salivary pH was evaluated using universal indicator papers in the 0–14 pH range, applied sublingually.

### 2.3. Saliva Sampling

The children were instructed to refrain from food consumption, drinking liquids, use of chewing gum, and oral hygiene measures for at least two hours before saliva collection. Subjects exhibiting intraoral lesions were considered ineligible for inclusion [37]. In order to standardize and minimize the diurnal variation of the evaluated parameters, saliva of all subjects was collected between 8 and 11 a.m. [38]. Saliva samples were collected in sterile polypropylene containers, prior to clinical examination, under the supervision of the examiner. While comfortably seated on the dental chair with the head slightly tilted in front, the child was advised to remain calm and refrain from excessive oro-facial movements, to avoid swallowing the saliva or speaking. Unstimulated whole saliva was deposited in containers using the passive drooling method [39], until a quantity of approximately 2 mL of saliva was collected. The saliva samples were stored at −80 °C, until further analyses. 

### 2.4. Total Protein Content

Total protein content was measured using the Pierce™ BCA Protein Assay Kit (Thermo Scientific, Landsmeer, The Netherlands) in 96-well polystyrene microplates (Greiner Bio-One, Alphen aan de Rijn, The Netherlands) as described previously [40]. Serial dilutions of bovine serum albumin (BSA, Merck, Amsterdam, The Netherlands) were used to generate a standard curve. Optical readouts were obtained using a microplate photometer (Multiskan™, Thermo Scientific, Landsmeer, The Netherlands). All saliva samples were analyzed in duplicate.

### 2.5. Protease Analysis

Salivary proteolytic activity was measured based on the cleavage of fluorescence resonance energy transfer (FRET) substrates as described previously [41]. In brief, 49 µL of saliva was incubated with 1 µL of 800 µM PEK-054 ([FITC]-NleKKKKVLPIQLNAATDK-[KDbc]) or PFU−089 ([FITC]-FR-[KDbc]). Fluorescence was recorded for 1 h at 37 °C using a fluorescence microplate reader (FLUOstar Galaxy, BMG Laboratories) at an excitation wavelength of 485 nm and an emission wavelength of 530 nm. Proteolytic activity was expressed as the increase in fluorescence per minute (F/min). All saliva samples were analyzed in duplicate.

### 2.6. Alpha-Amylase

Salivary α–amylase activity was determined with a colorimetric-based enzymatic activity assay using an amylase-specific substrate (2-chloro-4-nitrophenyl-α-d-maltotrioside, Sigma-Aldrich, Zwijndrecht, The Netherlands). To 10 μL saliva (diluted 1:50 with HPLC-grade water), 90 μL amylase substrate was added. The substrate is cleaved by α–amylase into 2-chloro-4-nitrophenol, a yellow compound. The formation of this degradation product was determined using a microplate photometer (Multiskan™, Thermo Scientific, Landsmeer, The Netherlands). The absorbance was measured for 15 min at 405 nm immediately after the addition of the amylase substrate. A reference with a known concentration of α–amylase (3 U) was included in each plate. Salivary α-amylase concentration was expressed as units per milliliter (U/mL). All measurements were performed in duplicate.

### 2.7. Matrix Metalloproteinase

Salivary levels of MMP-8 and MMP-9 were analyzed by combining Simplex kits containing specific antibodies for each protein considered, to create a multiplex panel that utilized Luminex xMAP (multianalyte profiling) technology for detection and quantification (ProcartaPlex Multiplex Immunoassay, Affymetrix, eBioscience, Thermofischer, Vienna, Austria). The assay was carried out based on the protocol provided by the manufacturer, and data interpretation was done by setting the following cut-offs: 900 pg/mL for MMP-8 and 600 pg/mL for MMP-9.

### 2.8. Data Analysis

Statistical analyses were performed using Stata/IC 16 (StataCorp. 2019. Stata Statistical Software: Release 16. College Station, TX, USA: StataCorp LLC), and *p*-values < 0.05 were considered statistically significant. Data distributions were expressed as means, standard deviations (SD), medians, quartiles, intervals, and percentages. Quantitative variables were tested for normal distribution using the Shapiro–Wilk test. Correlations between variables were explored using either Pearson (*r*) or Spearman (*r*_s_) correlations coefficients. For categorical measures, Pearson Chi-squared tests were used. Fisher exact test was used when the expected frequency of any cell in the table was <5.

## 3. Results

The mean (SD) age of the participants included in the study was 8.2 (±2.6) years (range 5 to 15). The study group consisted of 6 boys (27.27%) and 16 girls (72.73%). The mean (SD) salivary pH was 6.81 (±0.40). 

Table 1 shows the mean and standard deviation values for caries indices, calculated for the number of surfaces and teeth affected. The majority of the children had mixed dentition at the time of the examination (77.27%, *n* = 17), while primary dentition was found in two children (9.09%), and permanent dentition in three children (13.64%). The mean (SD) and median values for MCL were 8.32 (±4.60) and 10, respectively. The average (SD) percentage of teeth affected by untreated dental caries per subject was 35.98% (±20.06%). No statistically significant association was found between salivary pH and carious indices.

The total protein concentration and enzymatic salivary activity are presented in Table 2. With respect to the enzymatic activity, a marginal significant negative correlation coefficient was observed between the PFU-089/TPC ratio and the caries indices for permanent teeth (*r*
_= =_ 0.45, *p* = 0.054 for DMFT and *r_s_* = −0.44, *p* = 0.057 for DMFS, Spearman’s rank correlation coefficient). Furthermore, similar correlations were observed between the values of the PFU-089/TPC ratio and the decay components of the carious indices, DT (*r_s_* = −0.44, *p* = 0.06) and DS (*r_s_* = −0.43, *p* = 0.069, Spearman’s rank correlation coefficient). On the other hand, a statistically significant positive correlation was observed between the values of the PFU-089/TPC ratio and the fs component of dmfs (*r_s_* = 0.49, *p* = 0.048, Spearman’s rank correlation coefficient).

Salivary alpha-amylase (sAA) concentration was significantly higher in the saliva of children who experienced fewer carious lesions in deciduous teeth, registered as scores for dmft (*r_s_* = −0.54, *p* = 0.017) and dmfs (*r_s_* = −0.59, *p* = 0.006, Spearman’s rank correlation coefficient). A similar negative correlation was observed between the sAA and dt/ds components (*r_s_* = −0.53, *p* = 0.019, and *r_s_* = −0.56, *p* = 0.013, respectively, Spearman’s rank correlation coefficient). Regarding salivary MMP concentrations, significant positive correlations were observed between MMP-8 levels and values of dmft (*r* = 0.62, *p* = 0.004, Pearson’s correlation coefficient) and dmfs (*r_s_* = 0.61, *p* = 0.006, Spearman’s rank correlation coefficient). These positive correlations were shown to be mainly the result of the decay components dt (*r* = 0.63, *p* = 0.004, Pearson’s correlation coefficient) and ds (*r_s_* = 0.64, *p* = 0.003, Spearman’s rank correlation coefficient). Marginal positive correlations were registered between MMP-9 levels and the caries indices for primary teeth (*r* = 0.45, *p* = 0.05 for dmft, Pearson’s correlation coefficient; *r_s_* = 0.48, *p* = 0.039 for dmfs, Spearman’s rank correlation coefficient). Our findings revealed no significant relationship between the investigated salivary parameters and carious experience in permanent teeth (DMFT/DMFS scores or any of the indices’ components).

Considering the MCL values, they showed moderately negative correlations with sAA values (*r_s_* = −0.51, *p* = 0.01, Spearman’s rank correlation coefficient) and positive correlations with MMP-8 values (*r* = 0.66, *p* = 0.0008, Pearson’s correlation coefficient). Accordingly, the percentage of untreated decayed teeth correlated positively with MMP-8 values (*r* = 0.63, *p* = 0.002, Pearson’s correlation coefficient) and negatively with sAA values (*r_s_* =−0.55, *p* = 0.008, Spearman’s rank correlation coefficient). Marginal significance was retrieved after testing the associations of MCL values with PEK-054 values (*r* = 0.41, *p* = 0.056, Pearson’s correlation coefficient), PFU-089 values (*r* = −0.43, *p* = 0.056, Pearson’s correlation coefficient), and MMP-9 values (*r* = 0.42, *p* = 0.055, Pearson’s correlation coefficient), as well as between the percentage of untreated decayed teeth and PEK-054 values (*r* = 0.4, *p* = 0.065, Pearson’s correlation coefficient), the PEK-054/TPC ratio (*r* = 0.39, *p* = 0.074, Pearson’s correlation coefficient), and MMP-9 values (*r* = 0.39, *p* = 0.069, Pearson’s correlation coefficient). The most important correlations between MCL values and salivary assessments are illustrated in Figure 1.

## 4. Discussion

Saliva has been lately brought forward as a reservoir of valuable biomarkers with diagnostic potential [42]. Furthermore, due to its wide availability and the stress-free and non-invasive means of collection, salivary diagnosis has been considered an ideal diagnostic tool, especially for children [43]. Dental pathologies that lead to modifications of the oral environment can therefore potentially be evaluated by analyzing changes in salivary composition [44]. The present study provides novel information regarding the association of several salivary biomarkers, such as alpha-amylase, total protease activity (using PEK-054 and PFU-089 substrates), MMP-8 and -9 levels, with dental caries activity in a children population, making them potential diagnostic and prognostic tools in the field.

The results showed that untreated dental caries were strongly correlated with elevated MMP-8 and MMP-9 salivary levels, and a positive relationship was observed for lesions affecting deciduous teeth. Such correlations were not detected in decayed permanent teeth. The explanation might reside in the differences in tooth morphology and carious pathogenesis between deciduous and permanent teeth [8,9,10], as well as in the fact that permanent teeth did not have enough time to decay to the same extent, in the context of a mixed dentition. Therefore, evaluating carious activity as the number and percentage of untreated caries seems more appropriate and offers a global assessment considering that our unit of analysis is the patient, not individual teeth, taken independently.

Subsequently, in a study conducted on a group of subjects aged from 10 to 55 years, Nascimento et al. (2011) observed that salivary MMP activity was dependent on carious activity, with no significant differences between age groups, including children. Patients with active caries exhibited significantly increased salivary MMP activities compared to those with chronic lesions. Moreover, MMP activity positively correlated with lesion depth, with statistically significant differences in MMP activity between mostly superficial lesions and those with pulp exposure [31]. Furthermore, another study detected significantly higher MMP-9 levels in dentinal fluid samples in the presence of teeth affected by irreversible pulpitis, compared to samples from teeth subjected to deep cavity preparations, but with healthy pulp [45].

Kobus et al. (2019) found a significant positive correlation between MMP-8 level and the number of teeth affected by active carious lesions in children with mixed dentition, suffering from juvenile idiopathic arthritis. However, in healthy controls, no significant correlation was found between dental caries and salivary activity of MMP-8 or -9 [46]. Tjaderhane et al. (1998) compared MMP levels in untreated saliva and acid-activated saliva and observed that the gelatinolytic activity of MMP-9 was increased 18.5-fold, and the collagenolitic activity of MMP-8 was 3-fold at low pH levels. They performed transmission electron microscopic and scanning electron microscopic analyses on dentin samples and demonstrated that acid-activated salivary MMPs have a clear dentin-degrading effect [26]. Regarding salivary pH levels, our results, however, did not reveal any relationship between pH and enzymatic or carious activity.

Regarding sAA activity, conflicting results on a relationship between this enzyme and carious experience have been reported in the literature [47]. Our study found that children with decreased carious experience had higher sAA activity. This observation is consistent with the results of Mojarad et al. (2013), who found an inverse relationship between sAA levels and dental caries. They emphasized that there might be a bi-directional influence, since low levels of sAA may promote dental caries, but dental caries may also decrease sAA activity [48]. In addition, sAA had been shown to exert antibiofilm efficacy [49]. Another study evaluating preschoolers, showed that sAA activity was 1.7 times higher in children without carious lesions compared to children affected by ECC. [50]. A possible explanation for these observations might be that sAA, by binding to cariogenic bacteria, enhances the removal of these microorganisms and therefore decreases dental caries incidence [19].

Conversely, some other studies showed that sAA levels were significantly higher in older children and adolescents with active caries when compared to caries-resistant subjects [51,52,53]. In overweight adolescent girls, a positive correlation between sAA levels and dental caries for permanent dentition was observed [54]. Similarly, Singh et al. (2015) observed in a group of children aged 4 to 8 years that sAA levels in children experiencing active caries were higher compared to those in children without carious lesions [55]. Positive correlations between dental caries and sAA were also found in young adults [56].

Interestingly, by being integrated in the dental plaque and presenting a site for adherence of amylase-binding bacteria, sAA modulates bacterial colonization and indirectly promotes tooth demineralization. However, as described above, not all research groups found significant relationships between sAA activity and carious experience. These conflicting results might also be related to the fact that sAA can be increased due to exposure to a stressor, and its levels might be affected by both psychosocial and physical stimuli. Such stimuli might be the psychological stress of the anticipated dental treatment and/or the physical activity required to reach the dental office [57]

No associations were observed between TPC and dental caries, for none of the dmf/DMF indices evaluated in the present study. Similarly, in another study, salivary TPC did not differ between children with ECC and their caries-free controls [58]. Another study performed in children and adolescents showed higher salivary TPC in caries-active subjects compared to caries-free subjects, but the differences did not reach statistical significance [14]. Furthermore, in another publication, it was reported that salivary protein levels increased with age group [59]. In a study performed on an adult population, significantly higher MMP-8 levels and TPC were registered for subjects with manifest carious lesions compared to subjects not affected by caries [27]. In addition, increased TPC levels were reported in adult women with a history of caries and in those with active caries [13].

Regarding salivary total protease activity, although with a marginal significance, PEK-054 levels positively correlated with dental caries, while for PFU-089, a negative correlation was observed in the present study. Generic protease substrates, such as PEK-054 and PFU-089, lack specificity, not allowing the identification and differentiation of the source of the protease. Bacterial and human proteases present in saliva may influence each other and potentially modify MMP activity [60]. Increased salivary protease activity has been registered in caries-active patients compared to caries-free patients, subsequently to increased activity of MMPs and cathepsins [44]. Despite the fact that salivary total protease activity has been intensely studied in relation to periodontal disease [61], to our knowledge there are no studies in children subjects evaluating total protease activity, using PEK-054 and PFU-089 substrates, in correlation with dental caries.

A multitude of salivary components play a role in cariogenic bacterial colonization and metabolism, in a relationship of either synergy or antagonism. This study evaluated a limited fraction of the salivary enzymes and, considering the complex composition of saliva, it provides only a small image of a broader picture and does not consider possible confounding effects of the other salivary factors. Furthermore, comparing our results to other studies might be flawed by the fact that different research groups used different methods of saliva collection and analysis. It is also difficult to clearly state that MMP activity is detected in the saliva of caries-active patients solely due to their carious experience, since MMPs can originate from various sources such as gingival fluid, dentinal lymph, or saliva. 

The sample size of this study was relatively small, as COVID-19 pandemic restrictions limited recruitment possibilities. However, the presented results are important for further research because they offer an illustration of how various salivary enzymes might influence and thus be related to carious activity in children. Understanding dental caries pathogenesis provides valuable information on how we can prevent and stop the destructive process once it has initiated. Future longitudinal studies should be conducted in order to evaluate the variability of these biomolecules’ levels over time. Moreover, it is important to investigate whether different dental treatments influence the salivary enzymatic activity. In order to exclude possible cofactors that might have influenced our results, such as stress levels, oral hygiene, and dietary habits, studies evaluating dental caries in a biological context should also include evaluations of these factors. In this respect, the results of this study should be confirmed on a larger scale study.

## 5. Conclusions

Within the limitations of the study, we conclude that salivary alpha-amylase and MMPs are potential indicators for monitoring dental caries progression. Total protein content and total salivary protease activity seem less suitable for this due to their lack of specificity.

## Figures and Tables

**Figure 1 children-09-00343-f001:**
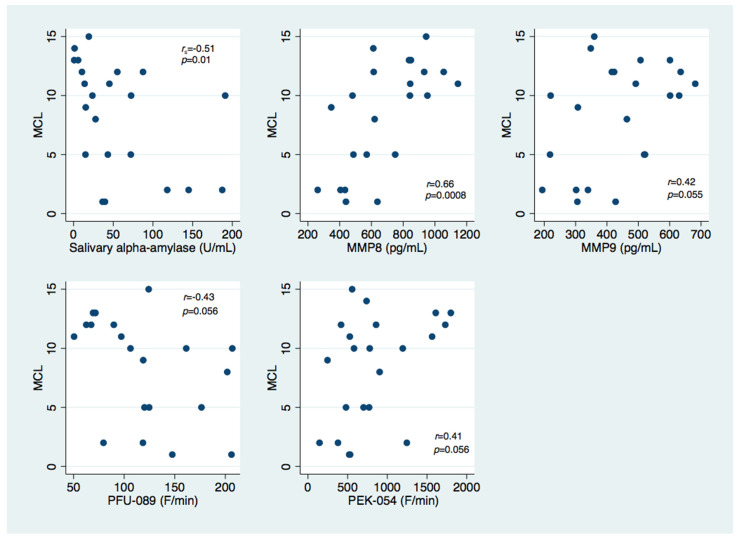
Scatter plot illustrating the relationship between MCL values and enzymatic activity levels for sAA, MMP-8, MMP-9, and total protease activity (using PFU-089 and PEK-054 substrates).

**Table 1 children-09-00343-t001:** Carious Experience in the Study Group.

**Indices for Primary Dentition (*n* = 19)**			
dmft	6.32 ± 4.57 (5)	dmfs	16.32 ± 14.89 (15)
d	5.95 ± 4.47 (5)	d	15.11 ± 14.79 (13)
m	0.16 ± 0.69 (0)	m	0.63 ± 2.75 (0)
f	0.21 ± 0.71 (0)	f	0.58 ± 2.09 (0)
**Indices for Permanent Dentition (*n* = 20)**			
DMFT	3.85 ± 4.02 (4)	DMFS	6.20 ± 7.76 (5)
D	3.50 ± 3.56 (3)	D	5.55 ± 6.50 (4)
M	0.05 ± 0.22 (0)	M	0.25 ± 1.12 (0)
F	0.30 ± 0.80 (0)	F	0.40 ± 1.19 (0)

d to children affected by ECC. AA, MMP-8, MMP-9, PFU-089 and PEK-054 m are ilustrated. Data are expressed as Mean ± SD (Median). dmft(s)/DMFT(S), Decayed Missing Filled Teeth (Surfaces); SD, standard deviation.

**Table 2 children-09-00343-t002:** Values of Salivary Proteins and Enzymatic Activity.

	Median	Q1	Q3	Mean	SD
sAA (U/mL)	38.04	15	72.49	55.74	57.17
TPC (μg/mL)	179.62	125.84	227.68	191.44	98.83
PFU-089 (F/min)	118.70	75.59	154.47	120.04	49.44
PFU-089/TPC ratio	0.70	0.37	1.47	0.91	0.69
PEK-054 (F/min)	719.61	519.73	1195.1	830.94	486.09
PEK-054/TPC ratio	4.35	2.72	7.10	5.26	3.54
MMP-8 (pg/mL)	629.09	480.68	848.87	684.54	244.91
MMP-9 (pg/mL)	425.23	307.26	521.37	432.42	145.19

sAA, salivary alpha-amylase; TPC, total protein content; MMP, matrix metalloproteinase; Q1, first quartile; Q3, third quartile; SD, standard deviation.

## Data Availability

The data presented in this study are available from the corresponding authors upon reasonable request.

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
