# Peer review of "Salivary Enzymatic Activity and Carious Experience in Children: A Cross-Sectional Study"

_children, 2022, doi:10.3390/children9030343_

Round 1

Reviewer 1 Report

The relevance of salivary constituents to protecting the teeth from caries or promoting caries is complex.  Although this study has a small sample size, and mixes both primary and permanent dentition, it still provides additional information that will be helpful for designing longer and larger studies. On line #268 of the manuscript, the verb should be "have" rather than "had".

Author Response

The relevance of salivary constituents to protecting the teeth from caries or promoting caries is complex.  Although this study has a small sample size, and mixes both primary and permanent dentition, it still provides additional information that will be helpful for designing longer and larger studies. On line #268 of the manuscript, the verb should be "have" rather than "had".

Thank you for your compliments! We have considered your suggestion and replaced the verb accordingly. The modification is visible in the attached manuscript, with track changes.

Reviewer 2 Report

The present research work shows the levels and activity of alpha-amylase and metalloproteinases 8 and 9 in children with caries.

The work in principle is interesting, although there are already some other previous reports about these molecules in caries.

The manuscript has some points to improve, mainly in the results section

  1. Table 1 is difficult to understand, it should be improved.
  2. If the most interesting results were regarding the correlations between molecules and caries, it is preferable to show these correlations a table or figure.
  3. The discussion is very long for the results obtained, as is the number of references.

Some other points to improve, but minor are listed below

  • Why didn’t the study include a caries-free group?
  • Why not use another diagnostic method for caries more modern, such as ICDAS?

Author Response

The present research work shows the levels and activity of alpha-amylase and metalloproteinases 8 and 9 in children with caries.

The work in principle is interesting, although there are already some other previous reports about these molecules in caries.

Thank you! We agree that the role of some of the molecules in caries development have been described previously. However to the best of our knowledge, no studies are available in which all enzymes (including PEK-054 and PFU-089) are studied. Below we will try to answer each of your suggestions. The changes we have made to the text in response to your suggestions are visible in the attached manuscript, with track changes.

The manuscript has some points to improve, mainly in the results section

  1. Table 1 is difficult to understand, it should be improved.

Following your suggestion, we have restructured Table 1. We hope that these changes will make Table 1 easier to understand.

  1. If the most interesting results were regarding the correlations between molecules and caries, it is preferable to show these correlations a table or figure.

Thank you for your suggestion. We have added Figure 1 in which we present the correlations between MCL values and each of the salivary parameters evaluated: sAA, MMP-8, MMP-9, PEK-054 and PFU-089.

  1. The discussion is very long for the results obtained, as is the number of references.

Considering your suggestion, we have removed some explanations regarding dental caries pathology and MMPs activity from the Discussion section (the respective paragraphs are highlighted with yellow). The respective information was adapted in order to be added to the Introduction section.

Some other points to improve, but minor are listed below

  • Why didn’t the study include a caries-free group?

In the Paediatric Dentistry Clinic where we conducted the present study, are treated mainly children who are affected by caries or affected by various other dental pathologies (including patients sent from private clinics in the area). Therefore, we were not able to recruit a sample of caries-free controls.

  • Why not use another diagnostic method for caries more modern, such as ICDAS?

Indeed, this is a good suggestion that we will take into consideration for future research. However, for the current investigation, we chose the WHO recommendations regarding carious indices calculation, due to its worldwide acceptance, convenience and the possibility to compare our results with previously published dental data.

Round 2

Reviewer 2 Report

I am pleased with the changes made to the manuscript and the responses of the authors.